# Inverted temperature gradients in gold–palladium antenna-reactor nanoparticles

Felix Stete [1], Shivani Kesarwani[1], Charlotte Ruhmlieb[2], Sven H. C. Askes [3], Florian Schulz [4], Matias Bargheer [1,5] ✉ & Holger Lange [1] ✉

In addition to enhanced fields and possible charge transfer, the concentration of photothermal energy at the nanoscale is a central feature of plasmon-driven photochemistry. It is well known that light energy can be efficiently concentrated in metal nanoparticles to length scales far below the wavelength of light. Here we demonstrate that the energy absorbed by a gold nanoparticle can be further localized within a bimetallic gold-palladium nanoparticle system by the dissipation of energy into the attached palladium satellite nanoparticles. After pulsed excitation of the gold core, the satellites collect nearly all photothermal energy and heat up by 180 K while the light-absorbing gold core remains much colder. By comparing transient absorption dynamics of a series of bimetallic nanoparticles with a three-temperature model, we can precisely assess the temperatures of the electronic and vibrational subsystems. We find a strong inverted temperature gradient that opposes the direction of energy input and concentrates the light energy at the active catalytic nanosite.

In photochemistry, metal nanoparticles are a versatile platform for harvesting optical energy, in particular via the large absorption cross section of plasmon resonances[1,2]. However, a problem of light-driven catalysis on plasmonic nanoparticles is the limited reactivity of the employed optically active metals. Most noble metals with a strong plasmon absorption in the visible regime are catalytically rather inactive, limiting their use to selected reactions. The most important metal catalysts are no photocatalysts, as their absorption cross section with visible light is low. As example, palladium (Pd) is among the most relevant transition metals for organic chemistry[3,4]. Example reactions are the oxidation of ethylene to acetaldehyde by air, the Wacker process, or the palladium-catalyzed carbon-carbon bond forming reactions developed by Heck, Negishi and Suzuki with many applications in target oriented synthesis, honored by the 2010 Nobel Prize in Chemistry[5,6]. Yet, also Palladium nanoparticles feature a low optical absorption. For this reason, bimetallic nanostructures of optically active noble metals and catalytically active metals have gained attention as potential photocatalysts[7–15]. From a fundamental point of view, three main processes are discussed in the context of plasmon-assisted chemistry: electric field enhancement, generation of non-thermal charge carriers and local heating[1,16–18]. Plasmonic nanoparticles have proven to efficiently transform optical energy to strongly localized heat, which is beneficial to overcome reaction barriers.[19,20] In this context, heat is often conceptualized as vibrational energy in the metals, reactants and solvents[21].

Here, we present a peculiar heating effect that provides even stronger energy localization and higher temperatures on catalytically active palladium satellites on spherical gold nanoparticles. We show that under pulsed excitation, almost all photoenergy is intermediately stored as heat in the palladium, resulting in temperatures far beyond

[1]Institut für Physik & Astronomie, Universität Potsdam, Potsdam, Germany. [2]Institut für Physikalische Chemie, Universität Hamburg, Hamburg, Germany. [3]Department of Physics and Astronomy, Vrije Universiteit Amsterdam, De Boelelaan 1081, Amsterdam, Netherlands. [4]Institut für Nanostruktur- und Festkörperphysik, Universität Hamburg, Hamburg, Germany. [5]Helmholtz Zentrum Berlin, Albert-Einstein-Str. 15, Berlin, Germany. ✉e-mail: bargheer@uni-potsdam.de; holger.lange@uni-potsdam.de

those in the gold particles although the latter have absorbed the main fraction of the light. Consequently, our results demonstrate that thermal energy in bimetallic systems can be efficiently localized in the catalytically active palladium, contributing a new perspective to the growing field of plasmonic antenna-reactor photocatalysis[22]

In plasmonic metals in general, optical energy is conterted to energy in the electron system, either by direct absorbtion into electron–hole pairs or via plasmon damping. Initially excited non-thermal electrons rapidly thermalize with the electron gas described by hot-electron Fermi distributions. In a direct-contact system of two metals, a natural assumption is a thermalized electron system in equilibrium throughout the heterostructure. In a subsequent step, the electrons dissipate energy to vibrations of the metal atoms via electron–phonon coupling. For a bimetallic system, the joint electron system interacts with the phonons of both materials via their specific electron–phonon coupling. If the electron–phonon couplings in the two metals differ significantly, interesting scenarios can develop, where light is absorbed in one metal, but the energy in the electron system is first dissipated in the other. Several recent publications that report modeling of material specific ultrafast X-ray diffraction data of nanolayered metal thin films have emphasized such unconventional phenomena[23–26]. The energy flow has also been discussed for bimetallic nanoparticles[27–30]. A faster energy dissipation was observed when gold nanoparticles were decorated with platinum. The studies do, however, not differentiate between phononic excitations inside the different metals and therefore do not assess local temperatures.

The phenomenon that we describe requires the combination of a metal with weak electron–phonon coupling (e.g., gold or copper) with a metal that exhibits a very large density of states at the Fermi-level (e.g., palladium, nickel or platinum), giving rise to strong electron–phonon interaction. We prepared a set of bimetallic gold-palladium nanoparticles with varying amounts of palladium satellites[31] and conducted transient absorption measurements at various excitation fluences. We have chosen palladium as reference material as robust synthesis protocols allow a control of the relative metal volumes and because palladium features the mentioned high catalytic activity. The transient absorption data are a measure of the electron temperature within the gold nanoparticles after photoexcitation[32]. A three-temperature model of the phonon temperatures of gold and palladium as well as the combined electron system reproduces all TA data with a fixed set of thermophysical parameters. We find substantial temperature gradients between the metals, with a temperature rise of 100-200 K within few ps in the catalytically active palladium, which exceeds the temperature rise in the gold by an order of magnitude for several tens of ps, although the two metals are in direct contact and only few nm large.

## Results

### Sample characterization

Our experiments rely on a robust synthesis method of the colloidal nanohybrids combined with a very careful characterization by transmission electron microscopy (TEM) and optical spectroscopy. Palladium-decorated gold nanoparticles were synthesized as described in the "Methods" section. The left column in Fig. 1a shows high resolution (HR-)TEM images of the bimetallic structures for the four different palladium loads. In all cases, small spherical satellites of palladium with a size of ≈2.5 nm formed around a gold core with a diameter of ≈21 nm. The distribution of the elements is confirmed in

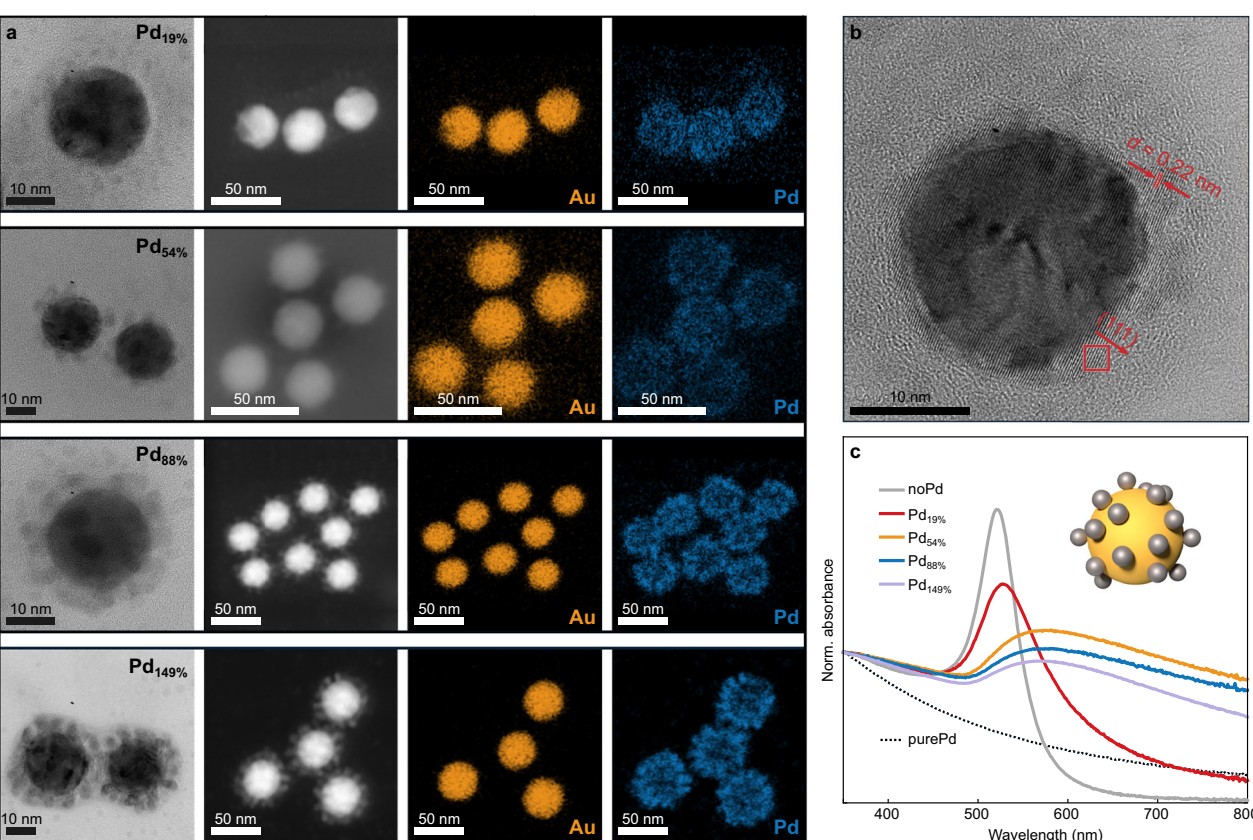

**Fig. 1 | Sample characterization. a** HR-TEM images (first column), TEM (second column) and respective EDX measurements of gold (third column) and palladium (fourth column) of the four samples containing palladium. The four images in one row belong to one sample (Pd$_{19\%}$, Pd$_{54\%}$, Pd$_{88\%}$, Pd$_{149\%}$ from top to bottom). **b** HR-TEM image of a Pd$_{19\%}$-particle. The crystal lattice of the palladium is clearly visible as indicated. Palladium grows epitaxially onto the gold particle without resolvable strain. **c** Absorbance spectra of the employed samples normalized to the absorbance at 350 nm (gray line: noPd; red line: Pd$_{19\%}$; orange line: Pd$_{54\%}$; blue line: Pd$_{88\%}$; purple line: Pd$_{149\%}$) and of pure palladium nanoparticles (gray dashed line). The inset shows a schematic representation of the bimetallic nanoparticle geometry.

**Table 1 | Summary of the different contents of palladium volumes in the different samples**

| Sample name | $V_{Au}$ ($10^3$ nm$^3$) | $V_{Pd}$ ($10^3$ nm$^3$) |
|---|---|---|
| noPd | 4.6 | 0 |
| Pd$_{19\%}$ | 4.8 | 0.93 |
| Pd$_{54\%}$ | 4.6 | 2.5 |
| Pd$_{88\%}$ | 4.7 | 4.1 |
| Pd$_{149\%}$ | 5.1 | 7.7 |

EDX measurements for gold (third column) and palladium (fourth column) corresponding to the TEM images presented in the second column. From these measurements, we retrieve the amounts of the two metals in the particles and their respective volumes. The results are summarized in Table 1, where the samples are named according to the ratio of the respective palladium and gold volumes. The high resolution micrograph in Fig. 1b reveals the crystalline nature of the palladium which is epitaxially grown onto the gold core in accordance with literature reports on similar structures[31,33,34]. The similarity between the lattice constants of gold and palladium leads to a quasi strain-free, direct interface[35]. The direct metallic contact is additionally confirmed by the absorbance spectra (Fig. 1c) of the colloidal solutions with increasing volume fractions of palladium. Already the smallest coverage of the gold sphere with palladium significantly broadens and shifts its plasmon resonance due to increased scattering[36]. For comparison, the spectra are normalized to the absorbance at 350 nm. Spectra of pure palladium nanoparticles show no significant resonance features in the visible and the broadening of the bimetallic particles is not only a spectral sum of the single components. The palladium satellites also directly absorb light in the field enhanced by the Au plasmon resonance. For small palladium fractions, the amount is significantly smaller than the amount of light absorbed in the gold.

To quantify this statement we evaluate the localization of the absorption in Pd$_{19\%}$ nanoparticles by optical simulations on a representative structure (Fig. 2). Integration of the absorbed power over the gold core and palladium satellites indicates that, at the experimental wavelength of 400 nm, the gold core is responsible for 84% of the total absorption. The simulated spectra qualitatively reproduce the redshift and damping of the LSPR according to the simulation packages COMSOL Multiphysics and Lumerical FDTD. This fact and further simulation details are given in the "methods" section on electromagnetic simulations.

### Transient absorption measurements

We recorded optical pump–probe spectra with the bimetallic particles excited with fs laser pulses at a wavelength of 400 nm at the interband transition in gold. As the focus of our study is on the thermalization dynamics between the electrons and the phonons, we did not choose a particular plasmon resonance. Possible differences in the electron thermalization pathways due to different excitations of the electron gas (inter- or intraband exitations) only play a role in the few tens of femtoseconds after excitation and are irrelevant after thermalization of the electron gas. Subsequent electron–phonon thermalization has been proven to be independent of the excitation pathway[37]. As the palladium satellites do not have a resonance in the visible spectrum and near UV, both the pump and the probe interaction that determine the transient spectra are dominated by the gold antenna (see Supplementary Note 4). The dominant absorption change originates from a shift and broadening of the plasmon resonance due to the heating of the electron gas[38]. As established previously[39], we determine the transient decrease in optical absorption (bleach) of the solution at the spectral position of the maximum change (see also Supplementary Note 5). It is shown as the relative increase in transmission $\Delta T/T_0$ normalized to the maximum relative change in Fig. 3a. The signal

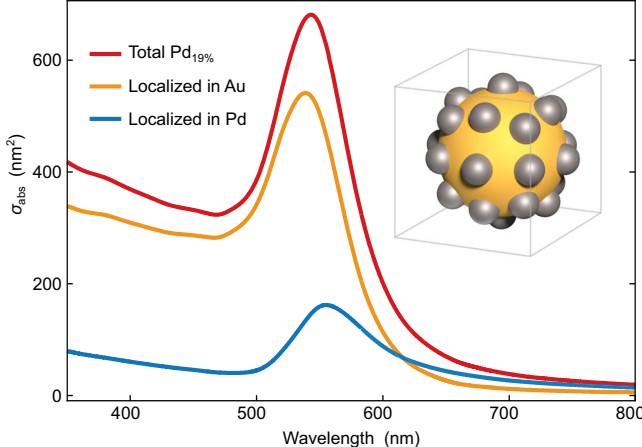

**Fig. 2 | Electromagnetic simulations simulations.** Simulated absorption cross section ($\sigma_{abs}$) of Pd$_{19\%}$ nanoparticles (red) separated into contributions to the absorption within the gold core (orange line) and within the palladium satellites (blue line). The inset shows the geometry of the Pd$_{19\%}$ nanoparticle in the Lumerical simulation environment. The gold core and palladium satellites have 10.5 nm and 2.5 nm radii, respectively. To account for the epitaxial growth of the palladium satellites onto the gold, the palladium nanoparticles were modeled as transected spheres in direct contact with the Au nanoparticle (cf. Fig. 1b).

slowly decays within about 6 ps for undecorated gold nanoparticles at high fluence. For increasing palladium volumes, we observe a faster decay of the signal. This change in transmission is a direct measure of the change in electron temperature[40]. This means that we directly observe a faster cooling of the electron temperature $\theta_e$ for increasing palladium loads (we refer to temperatures with the symbol $\theta$ to avoid confusion with the transmission $T$).

### Three-temperature model

To analyze the measured data, we set up a three-temperature model (3TM) which can determine the temperature evolution in both the free electron gas and the phonons in the respective metals. The 3TM describes the situation after electrons in the bimetallic structure are in thermal equilibrium. It is an established technique to model the material specific Bragg-diffraction thermometry in various bi-metallic heterostructures, where the phonon temperature of both materials can be simultaneously probed[23–26,41]. We assume that the electron gas has the same temperature $\theta_e$ in both metal components. It can exchange energy with the phonons in both gold and palladium via electron–phonon coupling with the respective constants $G_{Au}$ and $G_{Pd}$, resulting in a decrease of $\theta_e$ and an increase of the phonon temperatures $\theta_{ph,Au}$ and $\theta_{ph,Pd}$.

The temporal evolution of the three temperatures is given by the following coupled differential equations:

$$(\gamma_{Au}V_{Au} + \gamma_{Pd}V_{Pd})\theta_e \frac{\partial\theta_e}{\partial t} = -G_{Au}V_{Au}(\theta_e - \theta_{ph,Au}) - G_{Pd}V_{Pd}(\theta_e - \theta_{ph,Pd}), \quad (1)$$

$$C_{Au}V_{Au}\frac{\partial\theta_{ph,Au}}{\partial t} = G_{Au}V_{Au}(\theta_e - \theta_{ph,Au}), \quad (2)$$

$$C_{Pd}V_{Pd}\frac{\partial\theta_{ph,Pd}}{\partial t} = G_{Pd}V_{Pd}(\theta_e - \theta_{ph,Pd}). \quad (3)$$

Here, $C_{Au}$ and $C_{Pd}$ describe the heat capacities of the respective phonon systems. The heat capacity of the electron gas (($\gamma_{Au}V_{Au} + \gamma_{Pd}V_{Pd})\theta_e$) is given via the Sommerfeld constants $\gamma_{Au}$ and $\gamma_{Pd}$, which are proportional to the respective density of electronic states

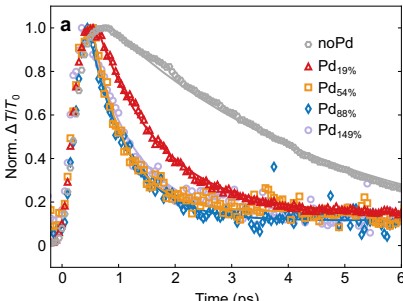
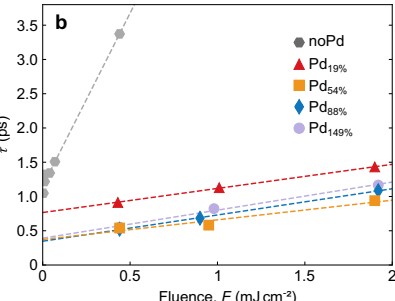
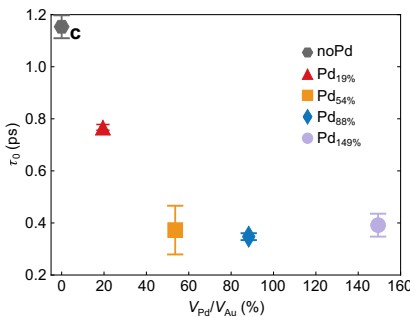

**Fig. 3 | Pump-probe dynamics. a** Transient absorption of hybrid particles with an increasing amount of palladium satellites for a fixed fluence of $F = 1\,\text{mJ cm}^{-2}$. The solid lines show exponential fits to the data. **b** Decay times for all samples under various pump powers and a linear fit to extract the effective decay time. **c** Extracted effective decay time for zero power excitation. In all panels gray represents the noPd sample, red the $Pd_{19\%}$ sample, orange the $Pd_{54\%}$ sample, blue the $Pd_{88\%}$ sample and purple the $Pd_{149\%}$ sample.

**Table 2 | Thermophysical parameters of gold and palladium**

|  | Au | Pd |
|---|---|---|
| $\gamma_i \left( \text{J m}^{-3}\,\text{K}^{-2} \right)$ | 71.5[56] | 400 |
| $G_i \left( 10^{16}\,\text{W m}^{-3}\,\text{K}^{-1} \right)$ | 2.1[57] | 40 |
| $C_i \left( 10^5\,\text{J m}^{-3}\,\text{K}^{-1} \right)$ | 24[58] | 29[43] |

Values without reference are optimized values for the simulation as discussed in the text.

DOS($E_F$) at the Fermi level[42]. Implicitly, we thus assume that the clean direct contact interface provides no relevant barrier for electrons between gold and palladium. The mismatch of the electronic band structures of gold and palladium does not impact the observed dynamics because at the Fermi velocity $v_F = 1.4\,10^6\,\text{m s}^{-1}$ of gold[42] the electrons traverse the gold nanoparticle within 14 fs and hence attempt many barrier crossings within the first hundreds of femtoseconds. Eqs. ((1)–(3)) neglect the direct heat exchange between the phonons of both metals. Although it can be highly relevant for the heat transport within the first two nm of the interface in epitaxial nanoscale metal thin films[23] it is less relevant here because the palladium particles have only small interface contact and the gold particle diameter is an order of magnitude larger. Hence, the coupling is dominated by the indirect coupling of the phonon systems via the combined electron system[25]. Introducing a direct coupling between palladium and gold phonons only slightly alters the temporal evolution of the temperatures. As its precise value depends on many unknown factors (exact particle morphology, lattice mismatch between the metals, etc.) it would therefore only introduce unnecessary additional parameters.

The 3TM describes the temperatures of electrons and phonons in the first picoseconds after optical excitation and thus sets in after the free electrons are in equilibrium. The detailed mechanisms of the plasmon damping are only relevant on the femtosecond timescale after excitation and therefore not of relevance here. The 3TM assumes that the energy absorbed from the laser pulse is distributed evenly between the free electrons. Solving this set of differential equations with the initial conditions of the temperatures after electron thermalization allows for determining the temperatures of the subsystems as a function of the time after optical excitation. The thermophysical parameters used in the modeling are given in Table 2. The remaining parameters, the volumes of gold and palladium, are experimentally determined from the EDX measurements and are listed in Table 1. Literature parameters already yield a very good agreement between the experiments and the model, however, for a perfect fit, we adjusted the Sommerfeld constant of palladium, lowering it from around $1000\,\text{J m}^{-3}\,\text{K}^{-2}$[43] to $400\,\text{J m}^{-3}\,\text{K}^{-2}$ and the electron–phonon coupling from $5.10^{17}\,\text{W m}^{-3}\,\text{K}^{-1}$[44,45] to $4.10^{17}\,\text{W m}^{-3}\,\text{K}^{-1}$[43] to $400\,\text{J m}^{-3}\,\text{K}^{-2}$ . We

rationalize the adjustment of the Sommerfeld constant and the electron–phonon coupling with the fact that the electronic system forms discrete particle-in-a-box states in ultrathin metal layers perpendicular to the surface. This - and additional interface effects - can modify DOS($E_F$) which is relevant to both values[42,46].

To cross-check the parameter set, we recorded fluence-dependent transients for all samples. In the limit of weak excitations, the temperature of the electrons can be described as $\theta_e = \theta_0 + \delta\theta$ and its evolution can be modeled by a single exponential $\theta_e(t) = \theta_{\text{end}} + (\theta_{\text{ex}} - \theta_{\text{end}})e^{-\frac{t}{\tau}}$. In this expression, $\theta_{\text{ex}}$ is the electron temperature after excitation and $\tau$ is the electron–phonon coupling time in the limit of weak excitation[47]. This electron–phonon coupling time is fluence dependent because the heat capacity of the combined electron system is proportional to the electron temperature $\theta_e$ (see Eq. (1)).

We extract $\tau$ for each measurement from the respective exponential fit as presented in Fig. 3b. Due to the low Sommerfeld constant of gold, the temperature of the electron gas in pure gold particles exceeds this linear approximation for the two largest fluences. This is why we added data at lower fluences in Fig. 3b. For particles consisting of only one metal, this decay time is given by the ratio of the Sommerfeld and electron–phonon coupling constants[27,48], which are both proportional to the DOS($E_F$) or the effective mass $m^*$ of the conduction electrons[46].

$$\tau = \frac{\gamma_{\text{Au, Pd}}}{G_{\text{Au, Pd}}}(\theta_{\text{init}} + \Delta\theta_{\text{ex}}). \tag{4}$$

Here, $\theta_{\text{init}}$ describes the electron temperature before optical excitation and $\Delta\theta_{\text{ex}}$ the increase of the electron temperature induced by the pump laser. The index gold stands for the situation of a pure gold particle, palladium for the case of a pure palladium particle. For hybrids with higher fractions of palladium, both the specific heat $\gamma_{\text{Au}}V_{\text{Au}} \ll \gamma_{\text{Pd}}V_{\text{Pd}}$ and the electron phonon coupling $G_{\text{Au}}V_{\text{Au}} \ll G_{\text{Pd}}V_{\text{Pd}}$ are dominated by palladium, and the gold contribution can be neglected. Figure 3b shows $\tau$ as a function of the fluence, i.e., the deposited energy $\Delta Q$ which is directly connected to the temperature change $\Delta\theta_{\text{ex}} = \sqrt{\frac{2\Delta Q}{\gamma_{\text{Au, Pd}}V_{\text{Au}}} + \theta_{\text{e, init}}^2} - \theta_{\text{e, init}}$. For sufficiently small $\Delta Q$, we therefore find from a Taylor expansion of $\Delta\theta_{\text{ex}}$ the linear expression:

$$\tau = \frac{\gamma_{\text{Au, Pd}}\theta_{\text{e, init}}}{G_{\text{Au, Pd}}} + \frac{\Delta Q}{G_{\text{Au, Pd}}V_{\text{Au, Pd}}\theta_{\text{e, init}}}. \tag{5}$$

For each colloidal sample, the deposited energy $\Delta Q$ had a slightly different proportionality due to differences in the absorption cross section at 400 nm and light extinction inside the solution due to varying particle density[47]. Therefore, the slopes of the linear fits in Fig. 3b do not allow for a direct extraction of the electron–phonon

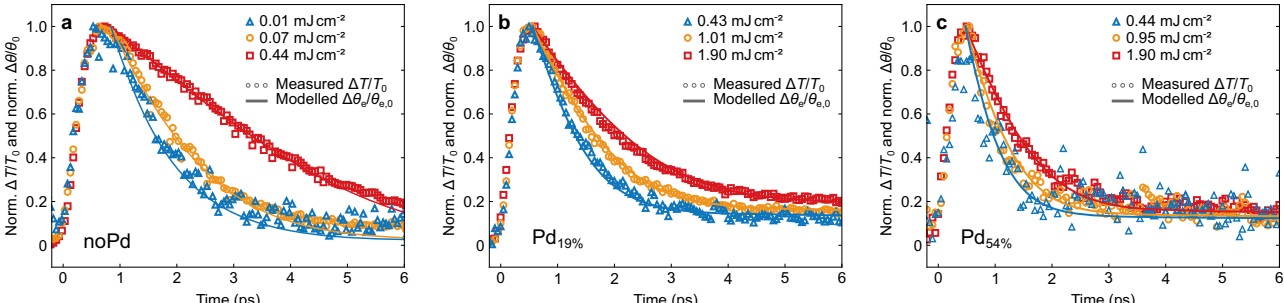

**Fig. 4 | Comparison of model and experiment.** Modeled changes in electron temperatures (solid lines) and measured relative transmission changes (markers) for the case of (**a**) pure gold particles, (**b**) a volumetric palladium to gold ratio of 19 % and (**c**) a ratio of 54 % for different excitation fluences as indicated in the legend (blue lines and markers represent the lowest, orange the medium and red the highest fluences). The model perfectly matches the experimental data.

coupling and the samples with higher palladium loads show a steeper slope when plotting over the incident - not absorbed - fluence. Yet, the intercepts with the $y$-axis yield the "zero-excitation electron–phonon coupling times" $\tau_0$ which are presented in Fig. 3c and directly deliver the ratio between $\gamma$ and $G$.

Already by this linearized analysis, we see how strongly the palladium influences the decay time in the particle and that already a small palladium volume fraction efficiently extracts energy from the electrons. The effective decay time $\tau_0$ extrapolated to zero fluence saturates at intermediate palladium volume fractions, because the electron system of palladium already dominates both the specific heat and the electron phonon coupling of the combined system. This fact is also reflected in the almost identical transients of the high palladium load samples in Fig. 3a. The limiting cases of the effective $\tau_0$ (no palladium and saturated $\tau_0$ in Fig. 3c) are in good agreement with the values in Table 2, supporting the model and the choice of the parameters.

We now use this set of parameters to model all transients by solving the 3TM. We obtain agreement between experiment and model for all fluences and palladium loads as presented exemplarily in Fig. 4. The plots for the two additional palladium loads are documented in Supplementary Note 6. The only fitting parameter for each sample is a proportionality constant relating the fluences to the actually absorbed pulse energy $\Delta Q$, as the absorption cross section is not precisely known and variations in the particle density in each solution can slightly alter the absorbed energy. However, this proportionality constant is kept fixed in each fluence series. We directly compare the measured transmission change $\Delta T/T_0(t)$ to the modeled temperature change of the electrons $\Delta\theta_e/\theta_{e,0}(t)$[27,39,40].

## Discussion

The agreement between the three temperature model and the full systematic set of measurements gives us confidence to interpret transient temperatures in all three subsystems, although only $\theta_e(t)$ is quantified by the measurement. Our interpretation of the model is strongly backed up by ultrafast X-ray diffraction experiments on bimetallic thin film samples, where both temperatures of the phonon systems are measured via their material specific Bragg peaks[23–26]. Note that the 3TM is also likely to explain bidirectional interaction in bi-plasmonic particles as a heating effect of a delocalized electron gas[30]. Figure 5a exemplarily depicts the transient temperatures for an excitation fluence of 1.01 mJ cm$^{-2}$ for the sample with the smallest non-vanishing palladium concentration (Pd$_{19\%}$). The short excursion of the electron temperature beyond 1000 K is ended by the rise of the palladium phonon temperature within about 2 ps. The temperatures of the electrons and the palladium phonons are in equilibrium around 460 K and significantly higher than in the gold phonons (310 K) for several tens of picoseconds. This means that the energy is almost completely stored inside the palladium not only because the phonon

temperature is higher, but also because the electronic heat capacity is dominated by palladium. For several tens of picoseconds, the palladium temperature rise $\Delta\theta_{ph,Pd}$ is about an order of magnitude larger than the temperature change $\Delta\theta_{ph,Au}$ of the gold phonon system. For a sample with low palladium load, the main fraction of the light is absorbed in the gold which gives rise to an inverted temperature gradient in respect to the excitation. This originates from the low volume fraction of palladium, which we confirmed by by numerical modeling described in Supplementary Note 7. Finally, all temperatures equilibrate at a temperature significantly below the maximum temperature of the palladium phonons. These values naturally depend on the volume fraction of palladium/gold, and thus, the temperature gradient can be further increased by a corresponding tailored synthesis that also optimizes the cooling of the gold antenna to the solvent.

The schematic in Fig. 5b illustrates the relevant steps in this concentration of energy at the catalytically active Pd sites. (i) The light energy is collected by the gold core via plasmon or interband excitation. (ii) The electron gas in the whole particle thermalizes. (iii) Rapid electron phonon coupling in palladium locally transfers the heat to the phonons of the palladium. This yields hot palladium nanoparticles in close proximity of the cold gold core. (iv) The electron and phonon systems within the particle thermalize within around 100 ps, before dissipating the heat to the solvent. This process of localized strong heating of the palladium has the potential to be a cornerstone in the emerging field of pulsed catalysis[49].

In conclusion, we presented a bimetallic antenna–reactor system that strongly localizes heat to few palladium satellites on a gold nanoparticle after pulsed optical excitation, which even inverts the temperature gradient with respect to the energy input. To confirm this, we analyzed the transient absorption in bimetallic gold–palladium nanoparticles with systematically varied fluence and palladium volume fraction. A three-temperature model with a reasonable and fixed set of thermophysical parameters consistently reproduces the time-dependent temperature of the combined palladium/gold electron system, which is observed as a transient transmission change. The modeling finds a strong temperature gradient between the palladium satellites and the gold core, which absorbs the photon energy but remains vibrationally cold for several tens of ps. These local temperature gradients exceed 150 K within a few nm and may be of use for the selective photocatalysis of reactions on palladium with a high reaction barrier. Gold very efficiently collects the light energy via interband absorption or enhanced plasmon resonances and funnels much more energy into palladium than direct absorption of palladium nanoparticles would achieve. We presume that these findings are generally applicable to various material combinations and geometries, and shall be continued to find the optimum geometry for thermal management at the nanoscale, both for continuous wave illumination[11] and for pulsed illumination exploiting the spatio-temporal non-equilibrium. Since we have essentially used literature parameters in the

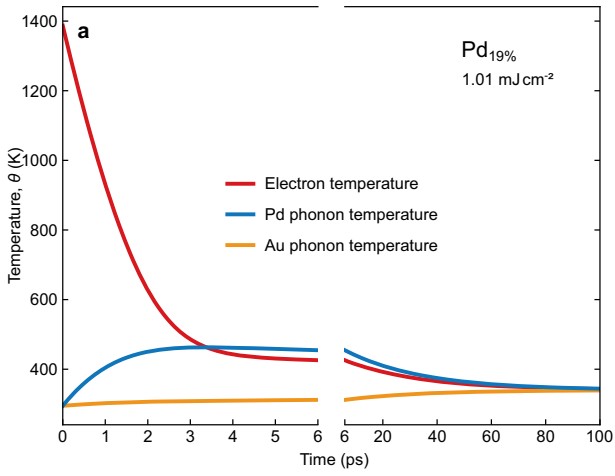

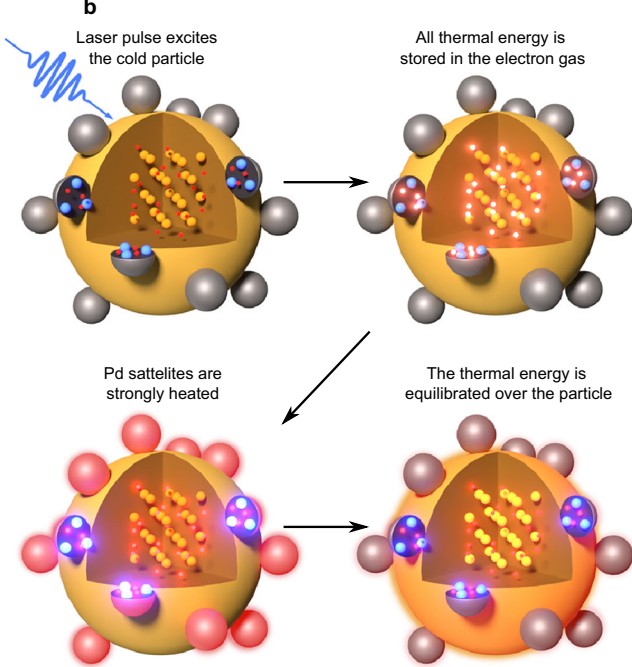

**Fig. 5 | Temperature gradient model. a** Temperatures of electrons (red), Pd phonons (blue) and Au phonons (orange) in the $Pd_{19\%}$ sample after excitation with a fluence of $1.0\ mJ\ cm^{-2}$. **b** Schematic representation of the effect. Electrons, gold ions and palladium ions are represented by red, orange and blue spheres, respectively. Heat in a subsystem is represented by a glow of the respective carriers of energy. The wavy blue arrow symbolizes the input of energy via the pump pulse.

modeling, the modeling has a predictive character and can be used to optimize the nanoscale hybrids as a photocatalyst.

## Methods

### Materials

Tetrachloroauric(III)acid ($HAuCL_4$, ≥99.9% trace metals basis), hexadecyltrimethylammonium bromide (CTAB, ≥98%, batch number 0000465936) hexadecyltrimethylammonium chloride (CTAC, ≥98%), L-ascorbic acid (AA, reagent grade), sodium borohydride ($NaBH_4$, ≥98%), hexadecylpyridinium chloride monohydrate (CPC ≥99%), sodium tetrachloropalladate(II) (≥99.9% trace metals basis), 1-octadecene (≥95%), palladium(II) acetylacetonate (≥99.9%), morpholine borane (≥95%) and oleylamine (≥98%) were purchased from Sigma Aldrich. All reagents were used without further treatment.

### Synthesis

Gold nanoparticles of 21 nm were synthesized based on the protocol presented by Zheng et al.[50], which was scaled up 25 times for high gold nanoparticle concentrations. The entire synthesis was carried out using ultrapure water (18.2 MΩ cm).

**CTAB-stabilized clusters.** Initially, CTAB (200 mM, 5 mL) and $HAuCl_4$ (5 mL, 0.5 mM) were stirred at 27 °C for 5 min. After mixing, $NaBH_4$ (600 μL, 10 mM) was quickly injected into the mixture under rapid stirring (1000 rpm) and then the mixture was stirred at 400 rpm for 3 min. The solution turned brown and was left undisturbed for 3 h.

**10 nm seeds.** 500 μL of CTAB-stabilized cluster was stirred with CTAC (20 mL, 200 mM) for a few min. After that, AA (15 mL, 100 mM) was added into the reaction mixture at 600 rpm, followed by a one shot injection of $HAuCl_4$ (20 mL, 0.5 mM). After stirring for 15 min (400 rpm), the AuNPs were washed with water by centrifugation (20,000 × g) for three times and the pellets were redispersed in CTAC (10 mL, 20 mM).

**Growth step.** CTAC (50 mL, 100 mM) and the desired volume of gold nanoparticles (in 20 mM CTAC) seeds were mixed. The mixture was then stirred (400 rpm) in a water bath at 30 °C, AA (325 μL, 100 mM) was added and after 2 min the addition of $HAuCl_4$ (50 mL, 0.5 mM) with a syringe pump at 50 mL h⁻¹ was started. After the complete addition of the Au precursor, the mixture was stirred (400 rpm) for additional 10 min at 30 °C and washed with water two times by centrifugation (20,000 × g). The obtained particles were uniformly spherical and featured a narrow size distribution (see Supplementary Note 1) and were redispered in CTAC (30 mL, 5 mM) for stability.

**Synthesis of bimetallic nanoparticles.** The synthesis of the bimetallic hybrid nanostructures was adapted from Guo et al.[31] Gold nanoparticle (5 mL, 0.88 nM) were washed with water two times by centrifugation (20,000 × g) and redispersed in 1 mL water. Different volumes (25, 50, 100 and 200 μL respectively) of $Na_2PdCl_4$ (10 mM) and 1 mL of gold nanoparticle were added to CPC (10 mM, 20 mL) solution at 65 °C. The mixture was stirred at 300 rpm. AA (400 μL, 100 mM) was injected at 500 rpm. After 2 min of vigorous stirring, the obtained solution was stirred at 300 rpm for 30 min. The final product was centrifuged with water three times at 20,000 × g for 15 min and redispersed in CPC (5 mL, 5 mM) for further analysis.

**Synthesis of plain palladium nanoparticles.** Palladium nanoparticles of 4 nm diameter were synthesized based on the protocol presented by Jin et al.[51] First, a mixture of octadecene-1 (8 mL), oleylamine (10 mL) and palladium(II) acetylacetonate (0.1 g) was heated to 100 °C under continuous nitrogen flow until the precursor melted. Subsequently, 0.2 g of morpholine borane mixed with 2 mL of oleylamine was injected into the above solution. The reaction mixture was then heated to 130 °C under stirring. 20 min later, the nanoparticles were washed with ethanol followed by centrifugation at 20, 000 × g for 10 min. After washing, the particles were kept in hexane (5 mL).

### Characterization

We employed a combination of HR-TEM imaging and EDS analysis using a JEOL JEM-2200FS operating at 200 kV equipped with a JEOL JED-2300 analysis station. The particle diameters were obtained from HR-TEM images. EDS analysis was conducted to obtain the relative elemental composition of the bimetallic nanoparticles (exemplified in Supplementary Note 3). The results from the EDS analysis and the diameter determination are listed in Table 3. Gold and palladium both crystallize in the face-centered cubic structure with the volumes of $0.0697\ nm^3$ per unit cell for gold and $0.0575\ nm^3$ for palladium. We

**Table 3 | Summary of the varying palladium amounts in the different samples**

| Sample name | $d_{Au}$ (nm) | Au-to-Pd atomic ratio |
|---|---|---|
| noPd | 21.0 | 0 |
| $Pd_{19\%}$ | 20.9 | 1:0.23 |
| $Pd_{54\%}$ | 20.7 | 1:0.63 |
| $Pd_{88\%}$ | 20.7 | 1:1.04 |
| $Pd_{149\%}$ | 21.4 | 1:1.76 |

confirmed the crystallinity with XRD characterization (see Supplementary Note 2). From the values for the particle diameters, the EDS values and the unit cell sizes, we deduced the elementary volumes presented in Table 1 which we used in the three-temperature model.

### Transient absorption spectroscopy

The transient measurements were conducted in a self built pump–probe set-up. 800 nm laser pusles with an length of 140 fs were generated at a repetition rate of 5 kHz were generated by a Ti:sapphire laser system (MaiTai/Spitfire Pro by Spectra-Physics). The main portion of the light underwent frequency doubling and was focused onto the particle solution as the pump beam, with a focal spot size of around $300\,\mu m$. A minor fraction of the laser system's output was utilized to produce supercontinuum (white light) pulses using a 1 mm thick sapphire plate. Being guided into the entrance of an Avantes spectrometer, these pulses probed the sample's transmission with an adjustable delay relative to the pump excitation. The time delay $t$ between the pump and probe beams was controlled by a delay stage. The pump beam was modulated with a chopper at 125 Hz to measure the change in transmission $\Delta T$ between the pumped ($T_0 + \Delta T$) and unpumped ($T_0$) sample.

### Electromagnetic simulations

**Simulation geometry.** Optical absorption spectra were simulated using COMSOL Multiphysics 6.3 finite-element modeling (FEM) and Ansys Lumerical Finite-Difference Time Domain (FDTD) software (version 2023 R2). In both simulation packages, a representative geometry was built based on a spherical gold nanoparticle with radius $r = 10.5$ nm that was decorated with 26 hemi-spherical palladium nanoparticles ("satellites") with a diameter of 2.5 nm (see inset in Fig. 2), which replicate the epitaxial contact between Pd and Au observed in HRTEM, consistent with minimization of interfacial and surface energy during growth. Six of the satellites were located at coordinates $[x, y, z] = [r, 0, 0]$, $[-r, 0, 0]$, $[0, r, 0]$, $[0, -r, 0]$, $[0, 0, r]$, and $[0, 0, -r]$. Twelve were located at $[\pm r/\sqrt{2}, \pm r/\sqrt{2}, 0]$, $[0, \pm r/\sqrt{2}, \pm r/\sqrt{2}]$, and $[\pm r/\sqrt{2}, 0, \pm r/\sqrt{2}]$. The final eight were located at $[\pm r/\sqrt{3}, \pm r/\sqrt{3}, \pm r/\sqrt{3}]$. The total volume of the palladium nanoparticles was 19%.

**Software settings.** For COMSOL, the refractive index for both metals was used from Johnson and Christy[52] and for water from Daimon and Masumura (at 20 °C)[53]. The optical response was simulated using an "Electromagnetic Waves, Frequency Domain" domain between 350 and 800 nm with the full-field formulation. The particle was meshed with a physics-controlled mesh with a "Fine" element size. The convergence tolerances were physics-controlled and the standard solver was used. For Lumerical, the refractive index was used from Johnson and Christy for gold[52], and from Palik for palladium[54]. The background refractive index was set to that of water (1.333). The space around the nanoparticle was meshed at 0.25 nm with a buffer of 5 nm from the gold surface. The mesh refinement was set to "conformal variant 1". Symmetry was applied by setting the $x$ min boundary condition to antisymmetric and the $y$ min boundary condition to symmetric. The simulation stopped at an auto shutoff setting of $1 \times 10^{-8}$. A total-field scattered-field (TFSF) source was used with wavelengths between 300

and 800 nm. The absorption and scattering cross sections were retrieved using standard cross section boxes.

## Data availability

The data that support the findings of this study are available from Zenodo[55] and from the corresponding authors upon request.

## Code availability

The code used in this study is available from the corresponding authors upon request.

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

## Acknowledgements

F.St. and M.B. acknowledge funding by the Deutsche Forschungsgemeinschaft (DFG, German Research Foundation) - CRC/SFB 1636 - Project ID 510943930 - Project No. A01. S.K., F.Sch. and H.L. acknowledge funding by the federal cluster of excellence "Advanced Imaging of Matter" (EXC 2056, ID 390715994). S.H.C.A. acknowledges the European Research Council (ERC) for funding from the Horizon 2020 research and innovation program (Grant agreement No. 101117530). We thank Yannic Stächelin for his support with initial test measurements, and both him and Marc Herzog for their fruitful discussions.

## Author contributions

F.St. conducted and analyzed the pump–probe experiments and set up the three-temperature model. S.K. synthesized and characterized the nanoparticle samples with support from F.Sch., C.R. performed the TEM analysis, S.H.C.A. developed the FDTD model. M.B. and H.L. conceived the project and supervised the work. F.St. and M.B. wrote the initial draft and all authors contributed to writing the manuscript.

## Funding

## Competing interests

The authors declare no competing interests.
