## [Transparent Peer Review file · Nature Communications]

Inverted Temperature Gradients in Gold-Palladium Antenna-Reactor Nanoparticles

Corresponding Author: Professor Holger Lange

Version 0:

Reviewer comments:

Reviewer #1

(Remarks to the Author)

The Authors present a study of electron thermalization in bimetallic Au-Pd particle clusters. Their claim is that despite the dominant absorption of light in the Au antenna, electron-phonon coupling predominantly leads to deposition of energy in the Pd particles. This causes a large heating gradient with most of the temperature increase initially taking place in the Pd. Such work is of high interest in the field. However, the experiments are analyzed in a way that I find somewhat questionable and so suggest that the manuscript be revised. While the conclusions may be plausible, they are at the moment not justified enough.

- 1) Why do the Authors drive the system at 400 nm? The energy is very high and activates very many transitions. This is in conflict with typical studies on electron thermalization in metals.
- 2) Depending on the excitation energy, the evolution of hot carriers and their thermalization can be radically different. For reference, one can look at the excellent work on Sun et al. in Phys. Rev. B 50(20), 15337-15348 (1994) Femtosecond-tunable measurement of electron thermalization in gold. There exist different ways thermalization proceeds, as reported by Sun et al. and such effects need to be discussed. Otherwise, the fact that a three-temperature model fits the data is not conclusive. There maybe hidden competitive effects that are omitted.
- 3) How sure can one be that the system behaves the way as assumed? There are two distinct components from the material point of view, more if each Pd particles is counted separately. At least a verification that they behave as expected should be carried out. For Au this is done, but not for Pd.
- 4) How do the Authors convert the model predictions into changes of the signal? As far as I am aware, thermalization of hot carriers and its modelling has reliably been carried out only in gold and for other metals there are less data. At the moment I do not understand how the Authors convert the so-far-questionable (or so-far-unproven) three-temperature model into changes in transmission. What are the parameters and model that relate the three temperatures into permittivity and then an optical signal? Furthermore, the rate equations in eq. 1a-c only show the relations between the temperatures, but have no ties to the excitation signal (initial conditions, etc.)
- 5) Are the spectra in Figure 1c all in the same scale? Have they been somehow normalized so that at short wavelengths they are all of equal absorbance? How was absorption measured for Figure 1c? There seems to be no mention of how that was performed in the manuscript, nor how the data were processed.
- 6) The absorption spectra in Fig 1c show practically equal absorption in the range 350-450 nm or there about. I find it quite surprising that there is so little change in that range, especially since absorption in Pd increases with decreasing wavelength. So, I am curious as to why 150% of Pd (of the volume of Au) does not cause any increase in absorption below 450 nm, only above.
- 7) One of the fundamental assumptions in the manuscript is an equal temperature of the electron gas. How justified is that, in light of a complex, multi-particle system?
- 8) The Authors claim that the temperature gradient is inverse to the direction of the energy input. Is there any claim on where the energy of light is actually deposited? The Authors assume an equal temperature of the electron gas. Also, they drive the system at 400 nm, well away from the plasmonic peak of the Au antenna, so how can a claim be made that the energy is deposited in the Au? Also, plasmonic particles are known for field enhancements, so even if the particle was driven at the plasmon frequency of the Au, there can be significant enhancement of direct light absorption in the Pd. The more Pd there is, the more energy can be deposited directly in it.
- 9) The directly excited hot charge carriers can be preferentially created in specific parts of the bimetallic clusters, not necessarily in the Au nor uniformly. This would change the initial conditions for the evolution with eq. 1a-c and possibly

negate the assumptions/conclusions.

10) How many Au-Pd cluster samples in each case were investigated? Are the results an average of many realizations, or only one of each? Or only one, for which the transient spectra match the fitting and conclusions?

Reviewer #2

(Remarks to the Author)

This interesting study by Stete et al. deals with a three temperature modelling of ultrafast spectroscopic data on plasmon-enhanced interactions in colloidal bi-metallic Gold-core Palladium-satellite nanoparticles for enhanced catalytic activity. Under femtosecond optical pumping, the free electron gas generated via the plasmon-interaction in the Gold-core dissipates energy as heat by efficient electron-phonon interaction into the Palladium-satellites, where it can be used e.g. for catalytic reactions.

While the work shows some compelling results, some questions remain.

I cannot judge the spectroscopic methodology at the moment, since it is simply not mentioned in the SI.

For increasing the flow and readability of the manuscript, the authors should give information about their choice of the bimetallic system, WHY did they choose this combination specifically and what has been found in previous work (Gargiulo, Nat. Commun 2023, 14, 3813), which is cited but it is not mentioned further that core-shell structures (negatively affect) the heat generation and that the Gold-core Palladium-satellite interfaces seem to be important (which even plays in favour for the authors and their choice of bimetallic material combination). What do the authors expect in case of the formation of Gold-Palladium Janus particles and heat dissipation, they could be highly interesting for catalytic reactions?

Even though the described Gold-core and Palladium-satellite combination seems to be the sweet spot for the antenna-reactor discussion here, did the authors try bigger Palladium-satellite sizes? Or in other words, do the authors find the same trend as e.g. described by Fagan et. al. (ACS Nano 2021, 15, 1378) that the Gold-core size is rather unimportant for the electron-phonon coupling but the antenna/reactor size will probably have a big influence. Following this argument, what do the authors expect with increased/decreased Palladium-satellite sizes (additional experiments might be needed here)?

The significance of the presented work for the field is important. The work and characterization shown here supports the immediate results shown but I'm missing the general tunability of the results at the moment. The data analysis is sound as well as the interpretation.

I encourage resubmission and to re-evaluate after the main points have been fixed.

Reviewer #3

(Remarks to the Author)

In this study, the authors investigated the energy localization effect of bimetallic gold-palladium nanoparticles and demonstrated how pulsed light excitation concentrates heat on the palladium satellites, thereby affecting the structure catalytic performance. The changes in electron and phonon temperatures at different palladium concentrations were analyzed using the three-temperature model (3TM) and ultrafast X-ray diffraction experiments. The study systematically discusses how palladium doping affects the thermal behavior of the electronic system, especially the changes in electron decay time and energy transfer efficiency at different palladium amounts. The article is innovative, especially in the design of photocatalysis using bimetallic nanocatalysts. I recommend that, after addressing the following issues, this paper could be published in Nature Communications. Here are some points that the authors are suggested to pay attention to:

1. The authors showed a large temperature gradient between the gold core and palladium satellites but had not discussed in detail how this temperature difference affects catalytic activity. It is suggested that the authors further explore the relationship between temperature gradients and photocatalytic efficiency.
2. The introduction briefly introduces the application of gold and palladium in catalysis but does not fully explain why the combination of gold and palladium was particularly chosen. It is suggested that the authors further incorporate relevant studies from the literature to explain more on how bimetallic systems enhance catalytic activity and why gold-palladium is superior.
3. Although the 3TM model was used and compared with experimental data, the applicability of the model and the error analysis were not discussed in depth. It is suggested that the authors provide an evaluation of the model fitting errors and further explore the potential biases between the model and experimental data.
4. In Figure 2(b), at the same excitation power, why does the decay time first decrease and then increase with the increase of Pd? The manuscript does not explain this phenomenon using the three-temperature model (3TM).
5. The manuscript mentions the use of local heating by plasmonic resonance. However, the wavelength of the excitation light source (400nm) corresponding to the obtained experimental temperature data (Fig4a) is not at the absorption peak of the Au-Pd structure. Is there any data on the structure that is excited resonantly?

Version 1:

Reviewer comments:

Reviewer #1

(Remarks to the Author)

The Authors have answered most questions, though I would state that some of the answers are only superficial and do not really add anything to what was stated previously. While the results are indeed quite interesting and the three temperature model is capable of fitting the observed data, I do not yet agree with all the claims made in the manuscript. What it does do, undoubtedly, is explain the temperature evolution under the assumption of a single electronic temperature.

Specifically, at the moment I still question the assertion that absorption is dominant in the Au in the coupled system. Hence, I consider the "inverted gradient" as not yet proven. Due to interaction between an antenna and nearby satellites the electromagnetic energy may be redistributed between the interacting elements even if the initial transfer from an incident beam will be to the antenna. This is, after all, the basis of antenna-reactor systems. One can look at this in a coupled harmonic oscillator model and under appropriate parameters the "lossy reactor not well coupled to light" may absorb a lot more than is transferred to it directly from light. This happens through coupling where energy is first deposited as a plasmon in the Au nanoparticle and then is transferred into the Pd NPs and only there does it decay. I still think that this point is not justified.

Reviewer #3

(Remarks to the Author)

The authors addressed the comments from the reviewers in the first round of review in an appropriate way. I believe that the revised manuscript can be published.

Dear reviewers,

We thank all reviewers for their careful reading and the suggestions for improvement of the manuscript. We have updated the manuscript accordingly. In the following we give a point-by-point answer to each of the reports.

Best wishes,

Matias Bargheer and Holger Lange on behalf of the entire team.

Reviewer #1 (Remarks to the Author):

The Authors present a study of electron thermalization in bimetallic Au-Pd particle clusters. Their claim is that despite the dominant absorption of light in the Au antenna, electron-phonon coupling predominantly leads to deposition of energy in the Pd particles. This causes a large heating gradient with most of the temperature increase initially taking place in the Pd. Such work is of high interest in the field.

However, the experiments are analyzed in a way that I find somewhat questionable and so suggest that the manuscript be revised. While the conclusions may be plausible, they are at the moment not justified enough.

1) Why do the Authors drive the system at 400 nm? The energy is very high and activates very many transitions. This is in conflict with typical studies on electron thermalization in metals.

Reply: The photon energy indeed has an impact on the initial carrier dynamics, plasmon dephasing and electron-electron scattering, which occur well within the first 100 fs after excitation. The current consensus is that only the amount of absorbed energy is of relevance for the carrier cooling after electron-electron scattering. Any coherence or signatures of the excited transition are lost in the process. See e.g., Ref. 23, W.Y. Chiang et al., J. Phys. Chem. C 2023, 127, 21176: "Electron-Phonon Relaxation Dynamics of Hot Electrons in Gold Nanoparticles are Independent of Excitation Pathway" and our own work, Nano Lett. 2023, 23, 5943.

400 nm proved to be a useful excitation wavelength as the absorption in this regime is strong and dominated by gold interband transitions. This allowed us to set the excitation conditions such that the optical energy is absorbed well-defined within the gold. Exciting the plasmon resonance has the disadvantage, that the absorption in Au strongly depends on the peak broadening induced by Pd and sample inhomogeneity.

We have expanded the discussion of the excitation in the main text. (Page 3, Lines 137 ff)

2) Depending on the excitation energy, the evolution of hot carriers and their thermalization can be radically different. For reference, one can look at the excellent work on Sun et al. in Phys. Rev. B 50(20), 15337-15348 (1994) Femtosecond-tunable measurement of electron

thermalization in gold. There exist different ways thermalization proceeds, as reported by Sun et al. and such effects need to be discussed. Otherwise, the fact that a three-temperature model fits the data is not conclusive. There maybe hidden competitive effects that are omitted.

Reply: The paper by Sun, Vallée, Acioli, Ippen and Fujimoto represents the first of a series of seminal publications on femtosecond spectroscopy in Au films, which discuss interesting details of the electron-electron and electron-phonon equilibration. Fig. 12 in that publication (PRB 50, 15337 (1994)) shows the non-Fermi shape of the electron distribution and the relaxation towards a Fermi-distribution which cools towards the phonons. While such details are very interesting in many contexts, it is not of immediate relevance for our work. After the initial ultrafast non-equilibrium electron dynamics, signatures of the excitation pathway are lost. Our manuscript investigates the situation after the electron gas is thermalized and starts to equilibrate with the phonon systems. Therefore, in our model, the coupling to the phonon systems of the two materials is the key to the observation. The excellent simultaneous fit of all data with a single model and without adjustable parameters indicates that including more details of the early relaxation dynamics are not relevant for the observation. Electron thermalization – i.e. the modification of the electron distribution including deviations from hot Fermi distributions - have been directly observed in metal thin films by ultrafast photoemission (e.g. Kühne et al., Phys. Rev. Res. 4, 033239 (2022)).

We have expanded the discussion on electron cooling. (Page 3, Lines 137 ff)

3) How sure can one be that the system behaves the way as assumed? There are two distinct components from the material point of view, more if each Pd particles is counted separately. At least a verification that they behave as expected should be carried out. For Au this is done, but not for Pd.

Reply: We have conducted reference transient absorption measurement on Pd nanoparticles. The absorption is spectrally very broad and the transient absorption the signal is therefore extremely small and noisy, which confirms that the signals reported in the manuscript have a different origin.

By optical measurements, it is difficult to crosscheck the behavior of the Pd. However, we have previously used the TTM in thin film heterostructures with the same thermophysical properties. See our references [23-26] The material specific x-ray response quantifies the response of the different materials with agree and small electron-phonon coupling.

For very large Pd concentrations, the absorption within the Pd itself becomes dominant. However, since the metals are in contact they establish a common temperature extremely rapidly.

In the main text, we now discuss more carefully that for the pure gold sample and the lowest Pd concentration, the Au absorption definitely represents the dominant energy

input. (Page 6, Lines 344 ff) In the SI we have added the static and transient spectra of pure Pd particles (Supplementary Note 4).

4) How do the Authors convert the model predictions into changes of the signal? As far as I am aware, thermalization of hot carriers and its modelling has reliably been carried out only in gold and for other metals there are less data. At the moment I do not understand how the Authors convert the so-far-questionable (or so-far-unproven) three-temperature model into changes in transmission. What are the parameters and model that relate the three temperatures into permittivity and then an optical signal? Furthermore, the rate equations in eq. 1a-c only show the relations between the temperatures, but have no ties to the excitation signal (initial conditions, etc.)

Reply: Our 3TM starts when the electron system has established a Fermi distribution with a defined temperature, that only depends on the absorbed light energy, irrespective of the thermalization mechanism that would only modify observations in the first few hundred femtoseconds.

Because of the absence of reliable models for the optical response of such hybrid particles we use the widely employed simple linear response approach that relates the transmission change $\frac{\Delta T}{T} \propto \frac{\Delta \theta}{\theta}$ to the temperature change of the electron gas, as often done in the literature. (See our references 38,39)

We would like to highlight that the interpretation of the experiments in our paper in a 3TM rests on a large and solid body of experiments and theory surrounding the energy transport in thin-film heterostructures which use material specific ultrafast x-ray diffraction to simultaneously measure the temperatures within the heterostructures. See our references 23-26.

We rephrased a few sentences to highlight that the assumption of a three-temperature-model for metal-heterostructures with different electron-phonon coupling constants is a valid and well tested and natural strategy. (Page 3, lines 165 ff and Page 4, lines 212ff)

5) Are the spectra in Figure 1c all in the same scale? Have they been somehow normalized so that at short wavelengths they are all of equal absorbance? How was absorption measured for Figure 1c? There seems to be no mention of how that was performed in the manuscript, nor how the data were processed.

Reply: Thank you for spotting that missing description.

We added the statement that the absorbance plotted is normalized to the value at 350 nm in the caption of Fig. 1c. The absorption was measured in solution (see Page 5, line 279). For the transient absorption experiments, we worked with concentrations low enough to ensure that every particle sees the same photon density. The average interparticle distance is such that coupling effects are not of relevance and that the dynamics is thus governed by the individual dynamics within the particles.

6) *The absorption spectra in Fig 1c show practically equal absorption in the range 350-450 nm or there about. I find it quite surprising that there is so little change in that range, especially since absorption in Pd increases with decreasing wavelength. So, I am curious as to why 150% of Pd (of the volume of Au) does not cause any increase in absorption below 450 nm, only above.*

Reply: See the comment above. The signals are normalized. For large Pd concentrations the interband absorption in Au remains, the plasmon resonance is broadened, and an extremely broad and featureless absorption of Pd is superimposed. From the different slope of the pure Pd absorption spectrum we can confirm that even for the largest Pd concentration the absorption of Au is still relevant.

7) *One of the fundamental assumptions in the manuscript is an equal temperature of the electron gas. How justified is that, in light of a complex, multi-particle system?*

Reply: This is a well-justified assumption after electron-thermalization within the first 100 fs. As electrons propagate through the nanostructures at around Fermi velocity, many scattering events occur within 100 fs.

On Page 1, line 47 we added “rapidly” to the sentence that expresses that photo-excited non-Fermi distributions relax so fast that the exact time is not relevant to our model.

8) *The Authors claim that the temperature gradient is inverse to the direction of the energy input. Is there any claim on where the energy of light is actually deposited? The Authors assume an equal temperature of the electron gas. Also, they drive the system at 400 nm, well away from the plasmonic peak of the Au antenna, so how can a claim be made that the energy is deposited in the Au? Also, plasmonic particles are known for field enhancements, so even if the particle was driven at the plasmon frequency of the Au, there can be significant enhancement of direct light absorption in the Pd. The more Pd there is, the more energy can be deposited directly in it.*

Reply: Thank you for pointing this out. As mentioned in the answer to question 5, we added the extinction spectrum of a pure Pd nanoparticle solution to Fig 1c and discuss the absorption in Pd. We emphasize that for the smallest Pd concentration the concept of an inverted temperature gradient is indeed justified, because most energy is input via the Au absorption. This is also the most relevant configuration, because the antenna-reactor concept is most efficient for low Pd concentration

9) *The directly excited hot charge carriers can be preferentially created in specific parts of the bimetallic clusters, not necessarily in the Au nor uniformly. This would change the initial conditions for the evolution with eq. 1a-c and possibly negate the assumptions/conclusions.*

Reply: This is entirely correct and we hope that details of the initial conditions will be investigated in future studies. Yet, for our present study we do not think these processes are of relevance here. Please note that the observed physics are not about “hot” electrons that are produced in certain “hot spots” of plasmonic structures. We study the time scale after the free electron gas is in thermal equilibrium.

10) How many Au-Pd cluster samples in each case were investigated? Are the results an average of many realizations, or only one of each? Or only one, for which the transient spectra match the fitting and conclusions?

Reply: The systematic study was carried out with one colloidal sample for each volume fraction. Experiments at different fluence are carried out on the same samples. We did not suppress any data set and there was no need to tune the thermophysical parameters much.

Reviewer #2 (Remarks to the Author):

This interesting study by Stete et al. deals with a three temperature modelling of ultrafast spectroscopic data on plasmon-enhanced interactions in colloidal bi-metallic Gold-core Palladium-satellite nanoparticles for enhanced catalytic activity. Under femtosecond optical pumping, the free electron gas generated via the plasmon-interaction in the Gold-core dissipates energy as heat by efficient electron-phonon interaction into the Palladium-satellites, where it can be used e.g. for catalytic reactions.

While the work shows some compelling results, some questions remain.

I cannot judge the spectroscopic methodology at the moment, since it is simply not mentioned in the SI.

Reply: Thank you for pointing out this deficiency. We have added details of the transient spectroscopy set-up to the supplementary information.

For increasing the flow and readability of the manuscript, the authors should give information about their choice of the bimetallic system, WHY did they choose this combination specifically and what has been found in previous work (Gargiulo, Nat. Commun 2023, 14, 3813), which is cited but it is not mentioned further that core-shell structures (negatively affect) the heat generation and that the Gold-core Palladium-satellite interfaces seem to be important (which even plays in favour for the authors and their choice of bimetallic material combination). What do the authors expect in case of the formation of Gold-Palladium Janus particles and heat dissipation, they could be highly interesting for catalytic reactions?

Reply: We thank the reviewer for bringing up this point. The idea of studying Janus particles is very nice. They have been used to induce motion via light, which could now rely on temperature gradients at the nanoscale. For the applications in chemistry, we think that the larger surface of small Pd satellites is more helpful and we are driving towards gold nanorods decorated with Pd. Yet, we want to emphasize that the three temperature model does not make assumptions on the morphology as long as the particles are small (compared to the mean free electron path) and as long as electrical contact between all components is ensured. Thus, the model can also describe the thermal evolution of Janus particles.

We extended the introduction and describe the choice of the system. The findings of the Gargiulo paper are highlighted in the conclusion as the pathway to further optimize the bimetallic nano-catalysis. (Page 6, line 392ff)

Even though the described Gold-core and Palladium-satellite combination seems to be the sweet spot for the antenna-reactor discussion here, did the authors try bigger Palladium-satellite sizes? Or in other words, do the authors find the same trend as e.g. described by Fagan et. al. (ACS Nano 2021, 15, 1378) that the Gold-core size is rather unimportant for the electron-phonon coupling but the antenna/reactor size will probably have a big influence.

Following this argument, what do the authors expect with increased/decreased Palladium-satellite sizes (additional experiments might be needed here)?

Reply: For our largest Pd satellites, there is already 50% more Pd volume than Au, although in the TEM the “shell of particles” around Au is about 5 times thinner than the Au. Structures with much larger Pd spheres would essentially harvest the light in the Pd itself. We see that the thermalization timescale is already saturated for our large volume fractions, so that the reviewer’s question is rather a question about the electron-electron and electron-phonon scattering in pure Pd, which will certainly depend on the surface to volume ratio and on defects. In our story it is sufficient to see that in Pd the electron-phonon coupling is 20 times larger than in Au, and therefore the energy is localized in the Pd.

From the 3TM point of view, the size of the single Pd particles doesn't matter, only their amount (volume fraction relative to Au). From the perspective of possible catalytical applications, a larger Pd surface is likely to be beneficial. The Gargulio finding that satellites are different from shells, will be reconciled in diffusive 3TM that will become relevant when modeling the heat transfer to the solvent.

The significance of the presented work for the field is important. The work and characterization shown here supports the immediate results shown but I'm missing the general tunability of the results at the moment. The data analysis is sound as well as the interpretation.

I encourage resubmission and to re-evaluate after the main points have been fixed.

Reply: We thank the reviewer for highlighting the importance of the work. We would like to answer here, that the “tunability” is given from a theoretical perspective. Since our model uses literature values of thermophysical parameters, we can even say that the theory has predictive power, and this can be used for the rational design of highly effective nano-hybrids. We are working on this.

Reviewer #3 (Remarks to the Author):

In this study, the authors investigated the energy localization effect of bimetallic gold-palladium nanoparticles and demonstrated how pulsed light excitation concentrates heat on the palladium satellites, thereby affecting the structure catalytic performance. The changes in electron and phonon temperatures at different palladium concentrations were analyzed using the three-temperature model (3TM) and ultrafast X-ray diffraction experiments. The study systematically discusses how palladium doping affects the thermal behavior of the electronic system, especially the changes in electron decay time and energy transfer efficiency at different palladium amounts. The article is innovative, especially in the design of photocatalysis using bimetallic nanocatalysts. I recommend that, after addressing the following issues, this paper could be published in Nature Communications.

Here are some points that the authors are suggested to pay attention to:

1. The authors showed a large temperature gradient between the gold core and palladium satellites but had not discussed in detail how this temperature difference affects catalytic activity. It is suggested that the authors further explore the relationship between temperature gradients and photocatalytic efficiency.

Reply: We think that the fact that high temperatures are beneficial for driving chemical reactions is well known enough to just allude to it in the introduction. However, we now explicitly mention overcoming reaction barriers in the text. (page 1, line 29)

It is true that many more interesting things can be said about local temperature gradients, and indeed we plan to do experiments in the future that highlight gradients.

2. The introduction briefly introduces the application of gold and palladium in catalysis but does not fully explain why the combination of gold and palladium was particularly chosen. It is suggested that the authors further incorporate relevant studies from the literature to explain more on how bimetallic systems enhance catalytic activity and why gold-palladium is superior.

Reply: As mentioned in the answer to reviewer #1, we added a paragraph on the choice of the material combination. Palladium is a very relevant catalyst and gold the most relevant plasmonic metal. In addition, there are robust protocols to synthesize bimetallic Au-Pd particles. . (Page 1, line 9ff, page 2, line 79ff)

3. Although the 3TM model was used and compared with experimental data, the applicability of the model and the error analysis were not discussed in depth. It is suggested that the authors provide an evaluation of the model fitting errors and further explore the potential biases between the model and experimental data.

Reply: We added text reasoning the use of the 3TM based on previous studies and we added reference literature that suggests the linear-response approximation for transmission \sim temperature change is appropriate.

We have added error bars to Fig. 2c. We have improved the description in the manuscript that discussed the data presented in Fig 2b. (See Page Page 5, lines 279 ff). Here, it is important to understand that the measured timescales should depend on the absorbed fluence, while for technical reasons we must plot as a function of incident fluence. However, the data in Fig. 2c are the extrapolation to zero fluence, where this tiny difference vanishes by definition.

4. In Figure 2(b), at the same excitation power, why does the decay time first decrease and then increase with the increase of Pd? The manuscript does not explain this phenomenon using the three-temperature model (3TM).

Reply: We suppose that the reviewer mainly hints at the data points at 1.9 mJ/cm². We do not interpret the ordering of the decay time for the three highest Pd volume fractions. We say that the decay time saturates, as it approaches the bulk Pd value. Please read the answer to question 4: The unexpected ordering is probably a consequence of the slightly different absorbed fluence. It is not relevant, because we use the extrapolation to zero fluence.

5. The manuscript mentions the use of local heating by plasmonic resonance. However, the wavelength of the excitation light source (400nm) corresponding to the obtained experimental temperature data (Fig4a) is not at the absorption peak of the Au-Pd structure. Is there any data on the structure that is excited resonantly?

Reply: As discussed in reply to referee #1, the choice of the exciting-photon energy has no significant impact and the electron-phonon dynamics relevant for our system. We have conducted reference experiments with an excitation at the plasmon resonance that show the same dynamics (for the same absorbed amount of energy). However, the data is noisier and therefore we decided to carry out the fluence and volume fraction systematics with the reliable excitation at 400 nm, which also has the advantage that potential differences of the absorption of the pump pulses for the different samples are minimized. The plasmon resonance is strongly affected by the Pd.

REVIEWER REPLY

Reviewer #1 (Remarks to the Author):

The Authors have answered most questions, though I would state that some of the answers are only superficial and do not really add anything to what was stated previously. While the results are indeed quite interesting and the three temperature model is capable of fitting the observed data, I do not yet agree with all the claims made in the manuscript. What it does do, undoubtedly, is explain the temperature evolution under the assumption of a single electronic temperature.

- We thank the reviewer for agreeing on this central point of the manuscript and for assessing the results as interesting. Indeed, the response by many of our colleagues on international conferences was enthusiastic and we believe that our discussion provides an important new perspective for the field.

Specifically, at the moment I still question the assertion that absorption is dominant in the Au in the coupled system. Hence, I consider the "inverted gradient" as not yet proven. Due to interaction between an antenna and nearby satellites the electromagnetic energy may be redistributed between the interacting elements even if the initial transfer from an incident beam will be to the antenna. This is, after all, the basis of antenna-reactor systems. One can look at this in a coupled harmonic oscillator model and under appropriate parameters the "lossy reactor not well coupled to light" may absorb a lot more than is transferred to it directly from light. This happens through coupling where energy is first deposited as a plasmon in the Au nanoparticle and then is transferred into the Pd NPs and only there does it decay. I still think that this point is not justified.

- We thank the reviewer for pointing out that some readers may consider dominant absorption of Pd a possibility even for very low volume fraction. To prove our assumption correct, we set up a finite elements model in two separate numerical software packages: COMSOL and lumerical. The modelling shows that for our sample Pd_{19%}, 84% of the pulse energy are absorbed by the gold core (see also Figure below). We included these results in the Supplementary Information and added a reference in the main text: "While this statement [dominant absorption in Au] already follows from the low volume fraction of palladium, we corroborate the dominant absorption of light in gold by numerical modeling described in Supplementary Note 8". In the respective supplementary note, we then show the following figure and give details on the modelling.

We hope that this numerical analysis convinces the reviewer that all essential items of our description of the processes are solid.